# Haplotype of the astrocytic water channel AQP4 is associated with slow wave energy regulation in human NREM sleep

**Sara Marie Ulv Larsen**[1], **Hans-Peter Landolt**[2,3,4,5], **Wolfgang Berger**[3,4,6],
**Maiken Nedergaard**[7], **Gitte Moos Knudsen**[1,8], **Sebastian Camillo Holst**[1,5]*

**1** Neurobiology Research Unit, Copenhagen University Hospital, Rigshospitalet, Copenhagen, Denmark,
**2** Institute of Pharmacology & Toxicology, University of Zurich, Zurich, Switzerland, **3** Zurich Center for
Integrative Human Physiology, University of Zurich, Zurich, Switzerland, **4** Neuroscience Center Zurich,
University and ETH Zurich, Zurich, Switzerland, **5** Sleep & Health Zurich, University Center of Competence,
Zurich, Switzerland, **6** Institute of Medical Molecular Genetics, University of Zurich, Schlieren, Switzerland,
**7** Center for Translational Neuromedicine, University of Copenhagen, Copenhagen, Denmark, **8** Faculty of
Health and Medical Sciences, University of Copenhagen, Copenhagen, Denmark

* holst@nru.dk

University, St.Louis, UNITED STATES

**Data Availability Statement:** Individual
anonymized observations that underlie the data
summarized in the figures are included in S1–S5
Data files. For further information or details please

## Abstract

Cerebrospinal fluid (CSF) flow through the brain parenchyma is facilitated by the astrocytic
water channel aquaporin 4 (AQP4). Homeostatically regulated electroencephalographic
(EEG) slow waves are a hallmark of deep non–rapid eye movement (NREM) sleep and
have been implicated in the regulation of parenchymal CSF flow and brain clearance. The
human *AQP4* gene harbors several single nucleotide polymorphisms (SNPs) associated
with AQP4 expression, brain-water homeostasis, and neurodegenerative diseases. To
date, their role in sleep-wake regulation is unknown. To investigate whether functional vari-
ants in *AQP4* modulate human sleep, nocturnal EEG recordings and cognitive performance
were investigated in 123 healthy participants genotyped for a common eight-SNP *AQP4*-
haplotype. We show that this *AQP4*-haplotype is associated with distinct modulations of
NREM slow wave energy, strongest in early sleep and mirrored by changes in sleepiness
and reaction times during extended wakefulness. The study provides the first human evi-
dence for a link between AQP4, deep NREM sleep, and cognitive consequences of pro-
longed wakefulness.

## Introduction

Glial-dependent cerebrospinal fluid (CSF) flow through the brain parenchyma, by some
termed the glymphatic system, facilitates the removal of waste by generating a convective flow
of CSF and interstitial fluid [1,2]. Evidence suggests that the pathway relies on 3 main pro-
cesses: Firstly, bulk flow of CSF through perivascular spaces is generated by arterial pulsations
from the heartbeat and potentially to a lesser extent by pulsations from the respiration [3]. Sec-
ondly, movements of CSF from the perivascular space into the brain parenchyma relies on the
water channel aquaporin 4 (AQP4), which is highly expressed on astrocytic vascular endfeet

contact the Neurobiology Research Unit in Copenhagen (Dr. Holst: holst@nru.dk, or MD Ulv Larsen: sara.larsen@nru.dk) or the Psychopharmacology Laboratory at the University of Zurich (Prof. Landolt: landolt@pharma.uzh.ch).

**Funding:** This project has received funding from the European Union's Horizon 2020 research and innovation programme under the Marie Sklodowska-Curie grant agreement No 798131 (to SCH), by a fellowship from the Copenhagen University hospital, Rigshospitalet R151-A6534 (to SMUL) and by grants from the Swiss National Science Foundation 31-67060.01; 310000-120377; 3100A0-107874; 320030_135414; 320030_163439 (to HPL). 'The funders had no role in study design, data collection and analysis, decision to publish, or preparation of the manuscript.

**Competing interests:** The authors have declared that no competing interests exist.

**Abbreviations:** APOE, Apolipoprotein E; AQP4, aquaporin 4; CSF, cerebrospinal fluid; ECG, electrocardiogram; EEG, electroencephalographic; EMG, electromyogram; EOG, electrooculogram; HtMa, HtMi; LD, linkage disequilibrium; MAF, minor allele frequency; NREM, non–rapid eye movement; PVT, psychomotor vigilance test; SNP, single nucleotide polymorphism; SSS, Stanford sleepiness scale; SWE, slow wave energy.

[1,4]. Mice lacking AQP4 show a strong reduction in parenchymal CSF influx [4,5] and increased interstitial beta-amyloid depositions [6], which is ameliorated by sleep deprivation [7]. Thirdly, the inward flow of CSF through AQP4 channels mainly occurs during non–rapid eye movement (NREM) sleep [8], and in preclinical studies, glymphatic flow is positively correlated with slow wave intensity during different types of anesthesia [9].

Emerging data support the existence of sleep driven CSF movements and clearance in the human brain. Increased levels of intracerebral tau and ß-amyloid have been observed in healthy adults after sleep loss [10,11], and recently sleep dependent pulsations in the fourth ventricle were demonstrated in the human brain [12]. To date, however, no studies have described the link between *AQP4* and human sleep-wake regulation or investigated whether genetic modulation of AQP4 may have restorative effects on cognitive functions after sleep loss.

The gene encoding AQP4 is located on chromosome 18 (18q11.2–q12.1) [13] (Fig 1). Several thousand single nucleotide polymorphisms (SNPs) in the noncoding regions of *AQP4* have so far been identified, and their function(s) in the normal and diseased human brain is an active area of research. Human *AQP4* SNPs have been shown to impair cellular water permeability and water homeostasis in vitro [14]. Moreover, an array of studies have associated human *AQP4* SNPs with neurological disorders including Alzheimer's disease progression [15], vascular depression phenotype [16], leukariosis [17], outcome after traumatic head injury [18], edema formation [19,20], and the risk of stroke [21]. These findings suggest a link between AQP4 and the development of brain diseases associated with waste deposition and fluid movements. Recently, a single variant within *AQP4* was associated with a 15% to 20% change in AQP4 expression [22].

Here, we aimed to investigate the role of AQP4 in human sleep-wake regulation. We hypothesized that if NREM slow waves are the endogenous regulator of CSF brain pulsations, then a reduced expression of AQP4 should be associated with a compensatory increase in deep NREM sleep (Fig 1A). To investigate this association, a haplotype spanning *AQP4* was genotyped a priori in a previously collected sample of 123 healthy participants from controlled sleep deprivation studies. Data from all-night electroencephalographic (EEG) recordings in baseline and recovery sleep as well as measurements of subjective sleepiness and global alertness throughout 40 hours of prolonged wakefulness were analyzed (Fig 1E).

## Results and discussion

### The *AQP4*-gene harbors an 8-SNP haplotype associated with AQP4 expression

Initial examination of SNPs in the *AQP4*-gene revealed a conserved haplotype spanning the entire gene with 2 common variants (Fig 1B). This haplotype consists of 8 SNPs, including rs335929 implicated in cognitive decline in Alzheimer's patients [15] and rs162008 demonstrated to reduce AQP4 expression [22]. Based on overall variance in EEG slow wave energy (SWE; 0.75–4.5 Hz) and the a priori aim to detect an effect size of at least 5%, power analysis suggests a required sample size of 78 (39 per group) as sufficient (see Methods). The *AQP4*-haplotypes were compared by means of dominant analysis, whereby 45 heterozygous subjects and 7 homozygotes were denoted as carriers of the minor allele (HtMi) and compared to 71 individuals homozygous for the major allele (HtMa) variant (S1 Data). The 2 groups did not differ in demographic characteristics (S1 and S2 Tables), presented with similar sleep architecture, and had a normal response to sleep deprivation (S3 Table).

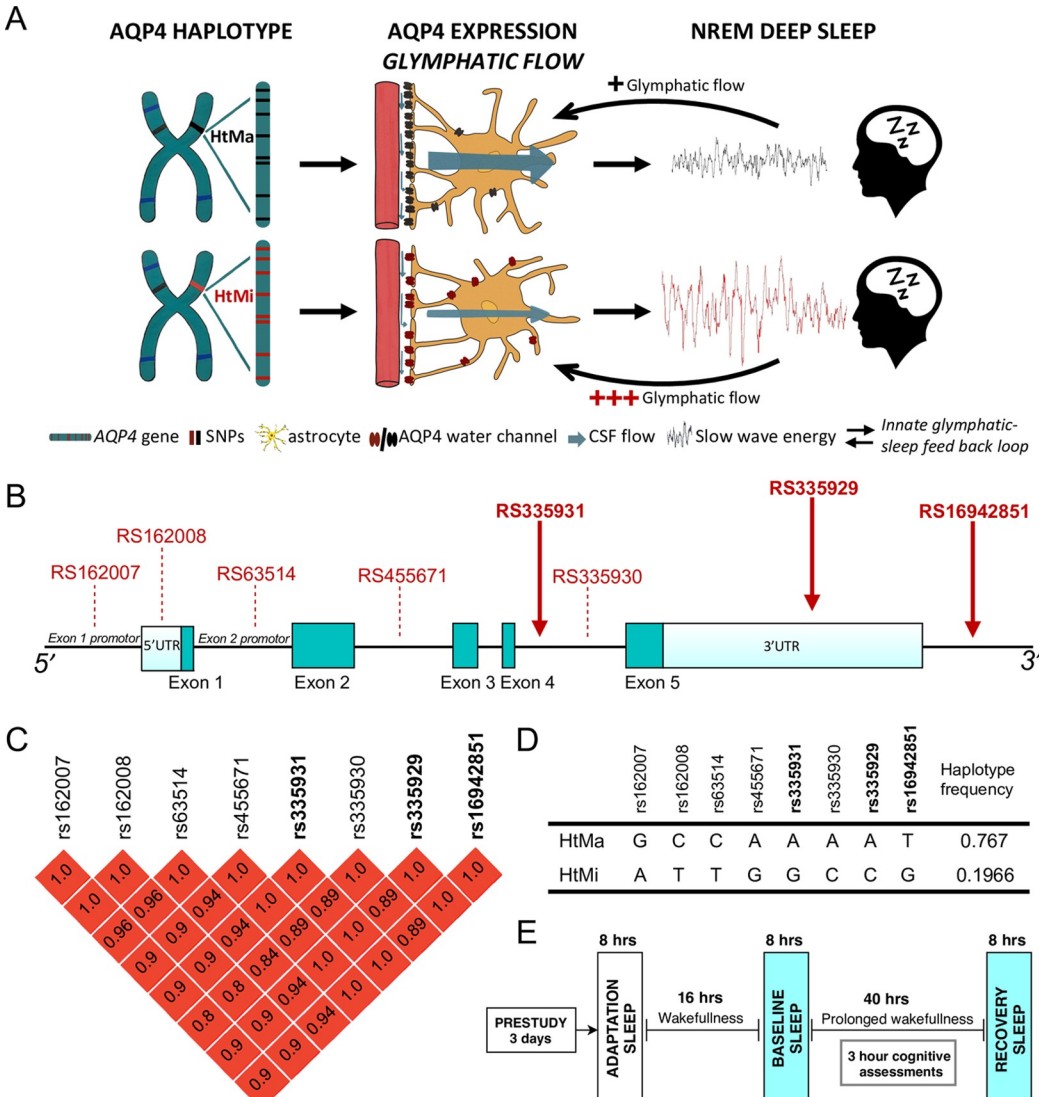

**Fig 1. Hypothesized role of the human *AQP4*-haplotype investigated in a controlled sleep deprivation study.** (A) Based on the genetic modulation of AQP4 protein expression [22], we hypothesized that the high AQP4 expressing variant of the *AQP4*-haplotype (HtMa; black) presents with improved glymphatic flow compared to the low AQP4 expressing HtMi variant (red). Assuming that NREM slow waves are the endogenous regulator of glymphatic flow, the HtMa variant require less SWE to initiate glymphatic flow than the HtMi variant, establishing an innate glymphatic-sleep feedback loop. (B) Physical map of the AQP4 gene and the location of the 8 haplotype SNPs. The 3 SNPs genotyped in this sample are marked with full red arrows. Dark green blocks: coding exons; light blue blocks: 5′- and 3′-untranslated regions. (C) LD block among the *AQP4* SNPs in the investigated haplotype. The pairwise LD coefficients ($r^2$) of SNPs in the LD block are color-scaled in red tones with dark red indicating perfect LD ($r^2 = 1$). (D) Table shows bases at the 8 different SNP locations in the AQP4 gene for the 2 haplotypes HtMa (76.7%) and HtMi (19.7%) and their respective frequencies in the CEU and TSI populations, representative of the investigated Swiss cohort. A total of 3.7% of CEU and TSI are predicted to be carriers of rare haplotype variants [23]. Frequencies in investigated study population were close to the prediction (HtMa: 75.7%; HtMi: 23.5; others: 0.7%; S1 Data). (E) Visualization of study design common for all subjects included from 6 separate studies. After a minimum 3-day inclusion period with monitored bedtimes and no caffeine intake, all study participants underwent an adaptation night in the laboratory before baseline sleep, 40 hours prolonged wakefulness and a recovery night, adding up to more than 1,950 hours ($123 \times 2 \times 8$ hours) of included sleep EEG recordings. Subjective sleepiness ratings and the approximately 10 min PVT were performed at 3-hour intervals. AQP4, aquaporin 4; CEU, Utah Residents from North and West Europe; EEG, electroencephalographic; HtMa, Major allele of haplotype; HtMi, Minor allele of haplotype; LD, linkage disequilibrium; NREM, non–rapid eye movement; PVT, psychomotor vigilance test; SNP, single nucleotide polymorphism; SWE, slow wave energy; TSI, Toscani in Italy;

## *AQP4*-haplotype is associated with a distinct modulation of slow waves in NREM sleep

The sleep EEG is genetically determined, with NREM sleep exhibiting up to 90% heritability [24], making it one of the most hereditary human traits described. To investigate whether the *AQP4*-haplotype modulates homeostatic sleep-wake regulation, EEG energy in predefined frequency bands in NREM sleep in baseline and recovery nights and the evolution of subjective sleepiness as well as cognitive performance measures were compared between the *AQP4*-haplotypes by a fixed sequence procedure [25] (see Methods). EEG SWE, which is a combined measure of sleep intensity and duration and is one of the best validated markers of sleep propensity in humans [26], was defined as the primary outcome variable. EEG quantification revealed that the HtMi carriers produced more SWE than the HtMa homozygotes ("genotype": $P < 0.03$; Fig 2A), an effect paralleled by post hoc slow wave power analysis (S2 Fig). These effects were not observed in the spindle range ($P > 0.77$) or any other frequency band ($P > 0.33$), nor in REM sleep (S1 Fig). The demonstrated *AQP4*-haplotype modulation of SWE documents an association between the intensity of deep NREM sleep and the expression of the AQP4-water channel, a relationship that may be central for CSF-driven brain pulsations. Given how recent preclinical evidence show that the intensity of slow waves is directly linked to glymphatic influx [9] and that the complete removal of AQP4 in mice results in brain impairments after sleep deprivation [7], this suggests an important role of AQP4-mediated clearance during sleep. The results appear in line with our initial hypothesis, suggesting that to compensate for a genetic reduction in AQP4 expression [22], AQP4 HtMi carriers increase SWE during NREM sleep, perhaps by an up-regulation of parenchymal CSF flow (Fig 1A). Interestingly, the *AQP4*-haplotype modulation was similar in baseline—and recovery nights ("genotype x night": $F_{1,120} = 0.37$; $P > 0.55$; $\eta_p^2 = 0.31\%$), proposing that the *AQP4* modulation is present both under normal sleep conditions and following the sleep homeostatic challenge.

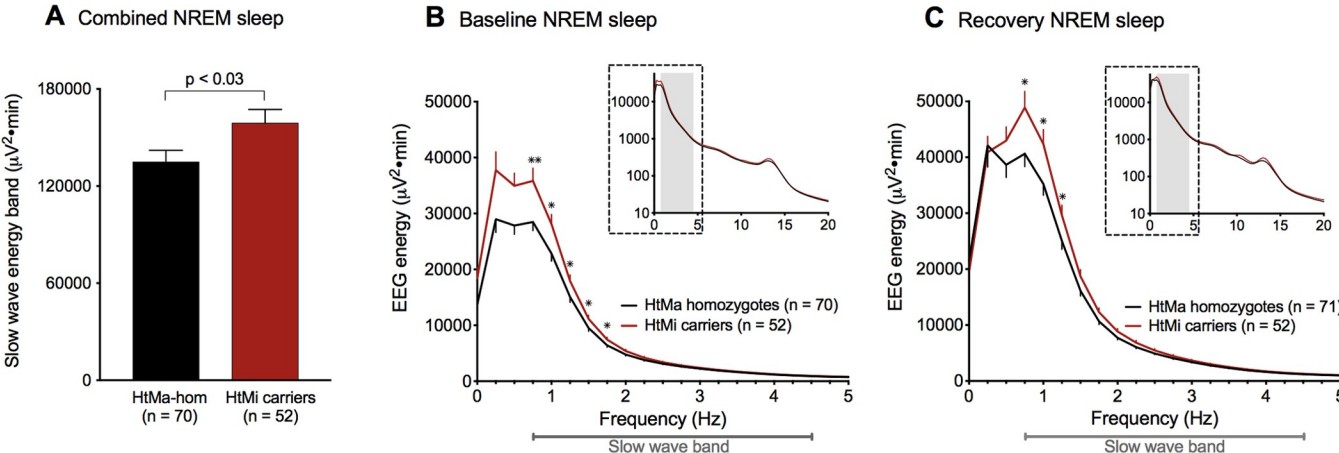

**Fig 2. *AQP4*-haplotype is associated with EEG energy regulation in the slow wave range.** Comparison of EEG energy (EEG power × time) across baseline and recovery nights in the slow wave range (0.75–4.5 Hz) within the *AQP4* haplotype variants HtMa homozygotes (black) and HtMi carriers (red). To minimize false positive results, EEG data was analyzed by a hypothesis-driven fixed sequence procedure that only revealed significant effects of AQP4 in the whole-night slow wave band, which was significantly increased in the *AQP4* HtMi-carrier group when compared to HtMa homozygotes (A; "genotype": $F_{1,121}$ = 5.0; $P < 0.03$; $\eta_p^2 = 3.95\%$). The effect was similar in baseline (B) and recovery (C) conditions and confined to the 0.75–2 Hz band. Inserts (B and C) represent full NREM sleep spectra for 0–20 Hz on log10 scale with gray shading indicating the slow wave band (S2 Data). Data represent mean ± SEM. By-bin unpaired two-tailed $t$ tests: $^*P < 0.05$, $^{**}P < 0.01$. AQP4, aquaporin 4; EEG, electroencephalographic; HtMa, Major allele of haplotype; HtMi, Minor allele of haplotype; NREM, non–rapid eye movement.

To localize the AQP4-dependent effect in the slow wave range, bin-wise frequency analysis was performed and revealed significantly higher energy in the 0.75 to 2 Hz range for the HtMi carriers than the HtMa homozygotes (Fig 2B and 2C and S2 and S5 Datas).

### EEG markers of sleep homeostasis in NREM sleep are associated with the *AQP4*-haplotype

To investigate whether the *AQP4* haplotype modulates the well-known homeostatic decline of slow waves across sleep, EEG SWE was quantified across the first 4 NREM sleep episodes in baseline and recovery nights. Consistent with the all-night sleep EEG analysis, a main effect of the *AQP4* haplotype was observed ("haplotype": $P < 0.03$; Fig 3A–3B), confirming the overall increased SWE levels in the HtMi-carrier group. Importantly, however, the haplotype effect was not constant across the night. Rather, *AQP4* HtMi carriers showed increased EEG energy mainly in the early part of the night ("haplotype x NREM episode": $P < 0.05$; Fig 3A–3B) when sleep pressure is highest (S3 Data). This finding may suggest an *AQP4*-dependent homeostatic modulation of NREM sleep. The effect in the first NREM episode was masked by a small delay in REM onset ($8.9 \pm 4.3$ min) in the HtMa homozygotes compared to the HtMi-carrier group ("haplotype": $F_{1,121} = 4.3$; $P < 0.05$; $\eta_p^2 = 3.41\%$; S3 Table), making the *AQP4*-haplotype modulation clearer when the first 2 NREM episodes were combined (Fig 3 inserts). The increased SWE in the low AQP4 protein expressing HtMi carriers indicates regulatory feedback between sleep intensity and AQP4 dependent water exchange (Fig 1A). Future studies are warranted to better understand whether the increase in SWE is also associated with increased CSF clearance or perhaps indicates that sleep intensity is related to an exchange of fluids across the blood brain barrier.

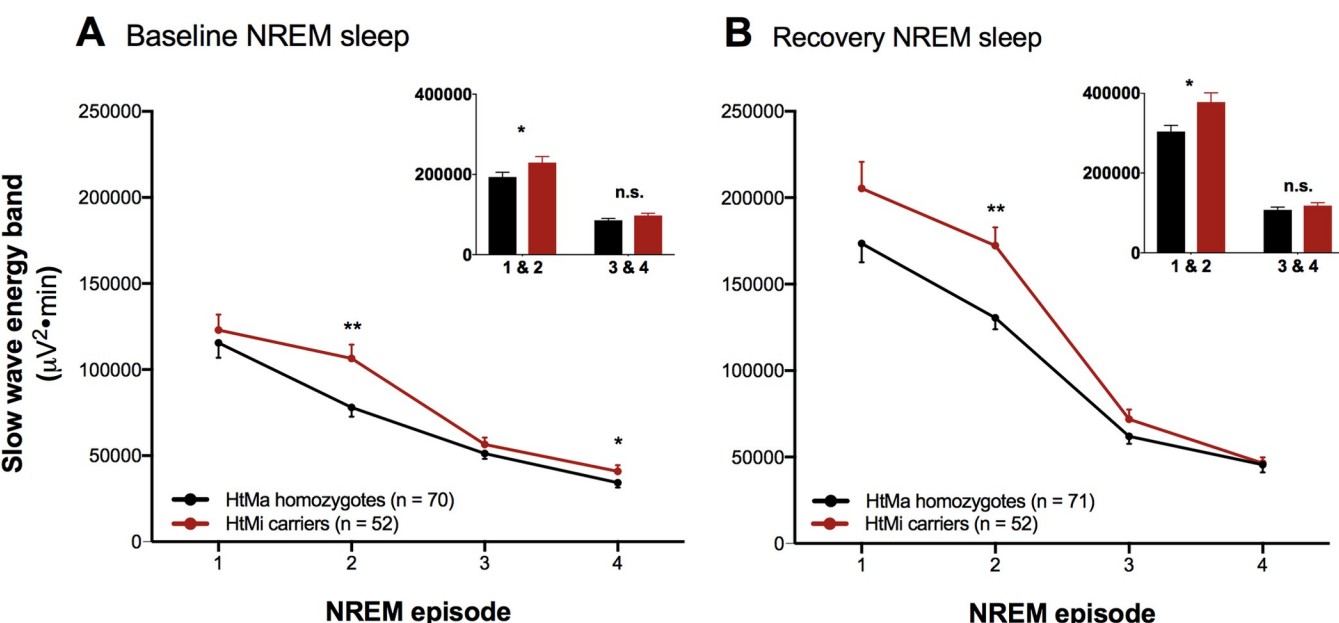

**Fig 3. Time course of EEG slow wave production is associated with the *AQP4* haplotype.** To probe the role of AQP4 on sleep-wake regulation, SWE across the first 4 NREM episodes (A and B) was investigated. The data confirmed the previously detected overall increase in SWE in the *AQP4* HtMi carriers (red) compared to the HtMa homozygote (black) group ("haplotype": $F_{1,121} = 5.39$; $P < 0.03$; $\eta_p^2 = 4.26\%$). Moreover, a significant AQP4-haplotype modulation across the first 4 NREM episodes was observed ("haplotype x NREM episode": $F_{3,843} = 2.65$; $P < 0.05$; $\eta_p^2 = 0.93\%$), an effect that was strongest in the second NREM episode. AQP4 HtMi carriers were found to have increased SWE mainly in the early part of the night (figure inserts). Spectral energy values of the slow wave band (0.75–4.5 Hz) in NREM sleep episodes 1 through 4 and the second part of the night (early: 1 and 2, late: 3 and 4) are plotted for the 2 haplotype groups for both baseline and recovery sleep (S3 Data). Data represent mean ± SEM. Unpaired two-tailed *t* tests: *$P < 0.05$, **$P < 0.01$. AQP4, aquaporin 4; EEG, electroencephalographic; HtMa, Major allele of haplotype; HtMi, Minor allele of haplotype; NREM, non–rapid eye movement; SWE, slow wave energy.

### The AQP4-haplotype is associated with a modulation of subjective and objective responses to prolonged wakefulness

To probe whether the *AQP4*-haplotype modulation of SWE has cognitive consequences, we investigated psychomotor vigilance performance (psychomotor vigilance test, PVT) and subjective sleepiness (Stanford sleepiness scale, SSS) across prolonged wakefulness in the 2 genetic groups (see Fig 1E). Subjective sleepiness ratings revealed that HtMi carriers coped slightly better with sleep deprivation than HtMa homozygotes and showed a smaller increase in sleepiness ratings from day 1 to day 2 ("haplotype x day": $P < 0.04$; Fig 4A). Importantly, median response speed on the PVT mirrored the effects on subjective sleepiness with the HtMi carriers reducing their speed slightly less than HtMa homozygotes ("haplotype x day": $P < 0.04$; Fig 4B), despite comparable speeds on day 1. Effects for lapses of attention were visually similar yet did not reach significance (Fig 4C and S4 Data). These data suggest that alterations in AQP4-expression have cognitive consequences, unveiled during sleep deprivation. Our data extend the recently described connection between slow waves, neuronal activation, and CSF flow in the fourth ventricle [12], by showing that the *AQP4* haplotype is associated with a modulation of NREM SWE and the restorative effect of sleep on cognitive functions. The low number of included HtMi homozygotes did not allow us to test whether the observed effects on sleep follow a dose response relationship, a question that should be addressed in future studies. Although we cannot rule out possible confounding associations between the *AQP4* haplotype and other sleep regulating genes, our observations may indirectly suggest that AQP4-dependent fluid flow within the neuropil is regulated by EEG slow waves during NREM sleep.

## Conclusion

Our data highlights that subjects carrying the low AQP4-expressing HtMi variant of the *AQP4*-haplotype have enhanced SWE mainly in early NREM episodes and cope slightly better with prolonged wakefulness. Given that SNPs associated with the HtMi variant affect the cognitive decline in Alzheimer's patients [15], the here described modulation of sleep intensity by the

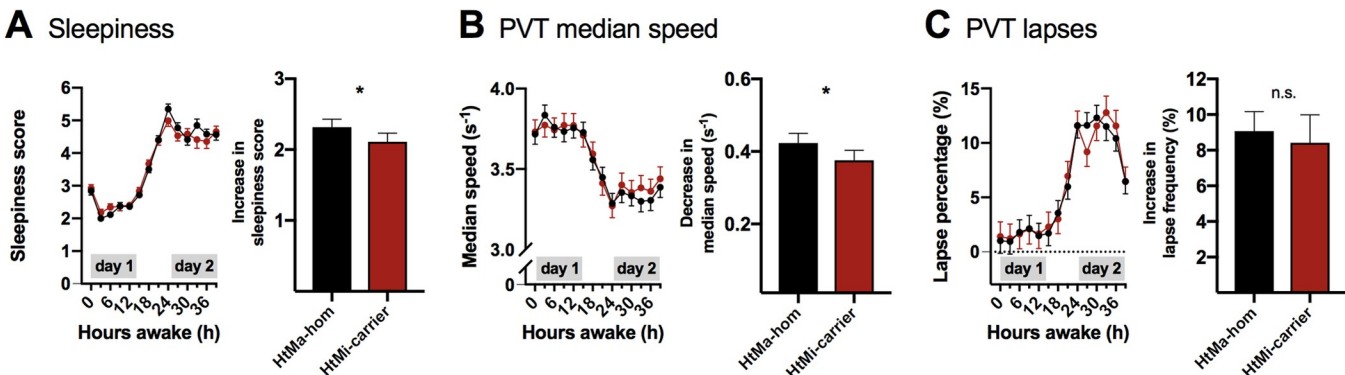

**Fig 4. Objective and subjective measures of sleep deprivation is affected by the *AQP4* haplotype.** Subjective sleepiness ratings on the SSS and objective alertness measures by the PVT. SSS scores (A), median speed (B), and attention lapses (C) on the PVT were quantified at 3-hour intervals across the 40 hours of prolonged wakefulness. Three-way linear mixed-model analysis revealed strong differences between day 1 and day 2 ("day": $F_{all} > 442$, $p < 0.0001$) and moderate modulations of clock time ($F_{all} > 4.6$, $p < 0.0005$) and by the "day x clock time" interaction ($F_{all} > 3.5$, $p < 0.004$) in all 3 measures. Comparison of subjective sleepiness between the 71 HtMa homozygotes (black) and the 52 HtMi carriers (red) of the *AQP4* haplotype revealed that the HtMi group coped slightly better with prolonged wakefulness than the HtMa homozygotes ("haplotype x day": $F_{1,364} = 4.5$; $P < 0.04$; $\eta_p^2 = 1.22\%$; Panel A right). Intriguingly, the *AQP4*-haplotype dependent modulation of sleepiness was mirrored by a similar effect on PVT median speed performance among the 60 HtMa homozygotes and 44 HtMi carriers that were tested ("haplotype x day": $F_{1,1041} = 4.7$; $P < 0.04$; $\eta_p^2 = 0.45\%$; Panel B right), an effect similar yet not significant for lapses of attention ("haplotype x day": $F_{1,1041} = 0.1$, $p > 0.78$, Panel C right; S4 Data). Data represent estimated mean ± SEM. *$p < 0.05$ from the corresponding "haplotype x day" interaction. AQP4, aquaporin 4; hom: Homozygote; HtMa, Major allele of haplotype; HtMi, Minor allele of haplotype; PVT, psychomotor vigilance test; SSS, Stanford sleepiness scale.

*AQP4* haplotype provides a tantalizing link between Alzheimer's disease, sleep intensity, and AQP4 in humans. Although future studies are warranted to better understand the underlying molecular mechanisms governing this effect, the data are consistent with the hypothesized impact from the glymphatic pathway. It also supports the hypothesis that sleep slow waves are part of the regulatory machinery of parenchymal CSF flow. Further studies investigating the *AQP4* haplotype and its association to sleep-associated brain functions are warranted.

## Methods

### Ethics statement

The study protocols were approved by the ethics committee of the Canton of Zurich for research on human subjects (Cantonal Ethics Committee reference numbers: E-39/2006 [29]; 2012–0398 [30]; E-24/2007 [31]; EK-Nr. 786 [32]; 2015–0424 [33]). As previously required, the oldest data set [28] was approved by local institutional review board. Subjects were recruited by public advertisements seeking participants for scientific studies or via billboard advertisements at the university buildings in Zurich. All subjects received a monetary compensation for their participation. Written and informed consent was obtained from all participants before the experiments as required according to the principles in the Declaration of Helsinki.

### SNPs of the AQP4 gene and haplotype analysis

To investigate genetic modulations of AQP4, common variants in the *AQP4* gene were explored using the dbSNP database build 152 (https://www.ncbi.nlm.nih.gov/snp/). The initial search in the 1000 genome project database within dbSNP revealed 32 SNPs with a global minor allele frequency (MAF) above 5%. Only 16 of these were common (MAF above 20%) in the European population data representative of the investigated Swiss cohort (CEU and TSI populations). Linkage disequilibrium (LD) analysis showed that 8 of the 16 SNPs (rs162007, rs162008, rs63514, rs455671, rs335931, rs335930, rs335929, and rs16942851) form a distinct haplotype with SNPs in high LD ($r^2 > 0.8$; Fig 1C). Further LD analysis of the haploblock revealed that these 8 SNPs form 2 common variants of the haplotype: a major haplotype (HtMa) with a 76,7% incidence and a minor haplotype (HtMi) with a 19.7% incidence (for further details, see Fig 1). Based on the 1000 genome database, this haplotype is widely detected across the European, American, and east Asian and south Asian populations.

### Genotyping of APQ4 SNPs

Genomic DNA extracted from 3 mL fresh EDTA-blood (wizard$^R$ Genomic DNA purification Kit, Promega, Madison, WI) was used for genotyping. The rs335931, rs335929, and rs16942851 polymorphisms of *AQP4* were chosen as tag SNPs to represent the haplotype and were determined using Taqman® SNP genotyping Assay (Life Technologies Europe B.V.; see also S2 Table). Allelic discrimination analysis was performed with SDS version 2.2.2 software (applied Biosystems, Foster City, CA). All genotypes were replicated at least once for independent confirmation.

The MAFs of the genotyped variants were in accordance with MAFs predicted by the dbSNP database (S2 Table). All 3 SNPs were in Hardy-Weinberger equilibrium. Pairwise LD coefficients ($r^2$) were calculated between rs335929, rs16942851, and rs335931 confirming high LD ($r^2 > 0,95$). The 2 haplotypes and allele frequencies are shown in Fig 1.

Because of the well-established association between Alzheimer disease risk and Apolipoprotein E (APOE) genotype [27], we checked the distribution of APOE genotypes (rs429358 and rs7412) among the *AQP4*-haplotype groups using the same Taqman® SNP genotyping

approach. The analysis revealed that the distribution was similar in HtMa homozygotes and HtMi carriers ($p > 0.47$; S1 Table).

## Study population

To examine the impact of the genetic haplotype of *AQP4* on the sleep EEG, we investigated data from 134 healthy participants of 6 previously published sleep deprivation studies [28–33]. All studies were conducted under strictly controlled conditions in the sleep lab of the Institute of Pharmacology and Toxicology at the University of Zürich, Switzerland, using similar protocols and methodology (Fig 1E). Two carriers of rare haplotypes as well as 9 older participants with an age above 60 years were excluded from the analysis. The total sample thereby included 71 individuals homozygous for the major allele (HtMa/HtMa), 45 heterozygous (HtMa/HtMi), and 7 homozygous for the minor allele (HtMi/HtMi). Given the low number of minor allele homozygotes, a HtMi-carrier group (HtMi/HtMi and HtMa/HtMi alleles) was created ($n = 52$). Dominant analysis was performed with the aim of investigating the consequence of harboring the minor *AQP4* haplotype. No difference in the distribution of the *AQP4* haplotype between the 6 studies was observed ($p > 0.21$). In studies that included the administration of one or more treatments [28–30], only data from the placebo arm were analyzed.

Study participants were right-handed healthy volunteers with a medical history free of neurological and psychiatric disorders. They were drug and medication abstinent and reported being good sleepers with regular bedtimes and no shift or night work. No participants passed through time zones or consumed excessive amounts of alcohol or caffeine in the 2 months prior to study enrollment. Before inclusion, participants underwent a screening night in the sleep laboratory to check for undiagnosed sleep disorders or low sleep efficiency ($<85\%$; see S1 and S3 Tables).

## Sleep study protocol

The 6 study protocols were very similar and were performed as follows: In the final 3 days leading up to the sleep studies, participants were required to keep a strict 8-hour/16-hour sleep schedule and to refrain from caffeine (coffee, tea, cola drinks, chocolate, and energy drinks) and alcohol intake. Compliance with these requirements was verified by actigraphy from a wrist activity monitor, sleep-wake diaries, and determination of saliva caffeine as well as breath alcohol levels upon arrival in the sleep lab.

The sleep studies consisted of a block of 4 consecutive nights (see Fig 1E): First and second nights served as adaptation and baseline nights, respectively. The subjects were then kept awake for 40 hours (i.e., for 2 days, skipping one night of sleep) until bedtime on the fourth night, when they were given a 10-hour sleep opportunity for recovery. During the period of prolonged wakefulness, the participants were constantly supervised by members of the research team and engaged in studying, playing games, watching films, and occasionally taking a walk outside the laboratory.

## Polysomnographic recordings

Continuous all-night polysomnographic recordings were performed on all baseline and recovery nights. The EEG, electrooculogram (EOG), submental electromyogram, (EMG) and electrocardiogram (ECG) were recorded using the polysomnographic amplifiers PSA24 (Braintronics Inc., Almere, the Netherlands; $n = 16$) [28] and Artisan® from Micromed (Mogliano Veneto, Italy; $n = 107$) [29–33]. In the recordings obtained with the PSA24 recording system, the analogue EEG signals were conditioned by a high-pass filter (3 dB at 0.16 Hz) and a low-pass filter (3 dB at 102 Hz), sampled at 512 Hz, digitally low-pass filtered (3 dB at 49 Hz), and stored with a resolution of 128 Hz. In the recordings obtained with the Artisan® recording system, analogue EEG data were conditioned with a high-pass filter (3 dB at 0.15

Hz), a low-pass filter (3 dB at 67.2 Hz), and sampled with a frequency 256 Hz. Sleep stages were visually scored in 20-second epochs according to standard criteria [34], and arousal- and movement-related artifacts were visually identified and removed. The data from the C3M2 derivation are reported. In both conditions, the analyses were restricted to the first 8 hours (480 min) after lights-off.

## EEG analyses

Four second EEG spectra (fast Fourier transform routine, Hanning Window, frequency resolution 0.25 Hz) were calculated with MATLAB (MathWorks Inc., Natick, MA), and EEG power spectra of 5 consecutive 4-second epochs were averaged and matched with the scored sleep stages.

The first 4 NREM episodes were defined according to current standards [35]. The all-night power spectra represent the average of all artifact-free 20-second values in NREM sleep (stages 1–4) between 0 and 20 Hz. The energy spectra contain all values of spectral power multiplied by time (minutes) spent in NREM sleep per night (480 minutes) or in the respective NREM sleep episodes. The energy calculations factors in a more relevant quantitative interpretation of the spectra [25].

## Cognitive testing and sleepiness ratings

The PVT is a simple reaction time task implemented in e-Prime software (Psychology Software Tools Inc., Pittsburgh, PA), in which subjects are instructed to press a button as quickly as possible with their right index finger when they see a digital millisecond counter that starts to scroll in the center of the computer screen [36]. Nineteen individuals were excluded from analysis because they performed a different, noncomputerized version of the task, resulting in a sample size of 104 subjects for the PVT task analyses. Moreover, because a subset of participants underwent neuroimaging on the second day of sleep deprivation, some performance measures are missing on day 2. Subjects received oral instructions and performed a training session prior to study start. For each PVT trial; 100 stimuli were presented (random interstimulus intervals: 2–10 s). Two extensively validated PVT variables were quantified [37,38]: "lapses of attention" (defined as the percentage of trials with reaction times longer than 500 ms) and median response speed (based on inverse reaction times). Immediately prior to all PVT assessments, a validated German version of the SSS was administered [39]. The sleepiness ratings of all 123 subjects were included in the analyses.

## Statistical analyses

All statistical analyses were performed with SAS 9.4 x64 software (SAS institute, Cary, North Carolina) and performed across nocturnal EEG data, subjective sleepiness and PVT performance measures. To approximate a normal distribution, EEG energy was log-transformed prior to statistical tests. Two- and three-way mixed-model analysis of variance were performed with the between-subjects factor "genotype" (HtMa homozygotes versus HtMi carriers) and the relevant within-subject factors: "condition" (baseline versus recovery), "NREM sleep episode" (1–4), "frequency bin" (bin 1–81), clock time (8, 11, 14, 17, 20, 23 o'clock), and the duration of prolonged wakefulness (day 1 versus day 2). For the nocturnal EEG data, 2 overall four-way mixed-model analysis were performed (for EEG power and EEG energy), which included all frequency bins, genotype, condition, and study ("bin x genotype x night": $P < 0.0001$). Further analysis was only performed because a significant effect of genotypes was established. Moreover, to control for type I error caused by multiple comparison, the significance levels for these primary 2 overall mixed models were set to $\alpha < 0.025$ (Bonferroni correction $\alpha = 0.05/2$). Type I errors were further controlled by only considering significant effects relevant when

the following 2 criteria were met: (A) more than 2 bins in the by-bin (mixed model) analysis of variance were below $\alpha$ = 0.05 and (B) the corresponding frequency band (slow wave/delta, theta, alpha, spindle, beta) was also significant at $\alpha$ level = 0.05. The final step implemented to control for type I errors was to investigate the sleep EEG bands using a hypothesis driven, fixed sequence procedure [25], in which EEG bands were listed based on their relevance for sleep-wake regulation as follows: (1) Delta/slow wave activity (0.75–4.5 Hz), (2) Spindles (12–15 Hz), (3) Theta (5–8 Hz), (4) Alpha (8–12 Hz), and (5) Beta (15–20 Hz) and tested in that order. Only if the previously tested band revealed a significant genotype modulation, statistical testing of the subsequent band was performed.

Based on the log-transformed EEG SWE data, a priori power analysis revealed that to determine a 5% difference between haplotypes with a simple two-tailed $t$ test ($\alpha$ = 0.05), a total sample size of 78 subjects (Cohen's d: 0.833; SWE mean: 5 log[$\mu V^2$min] ± 2.5%; standard deviation: 0.3 log[$\mu V^2$min]) are required (G*Power 3.1.9.2; Die Heinrich-Heine-Universität Düsseldorf).

When a significant main effect or interaction was discovered, appropriate paired or unpaired two-tailed $t$ tests were used to localize differences within and between groups. If not stated otherwise, only significant effects or results are reported. Effect sizes (partial eta squared: $\eta_p^2$) were calculated from corresponding mixed-model F-values and degrees of freedom. Effect sizes of 0.0099, 0.0588, and 0.1379 are considered small, moderate, and large, respectively [40,41].

## Supporting information

**S1 Fig. REM energy analysis.** REM energy analysis for *AQP4* HtMa homozygous (black lines) and HtMi carriers (red lines) across baseline (A) nor recovery (B) nights. No significant modulation by the *AQP4* haplotype was observed ("haplotype": $F_{1,21}$ = 0.07; $P$ > 0.79), confirming that the sleep EEG modulations are selective to the NREM slow wave range (S5 Data). Plots represent means; error bars represent SEM. (Insert) Full NREM sleep spectra for 0 to 20 Hz on log10 scale. AQP4, aquaporin 4; EEG, electroencephalographic; HtMa, Major allele of haplotype; HtMi, Minor allele of haplotype; NREM, non–rapid eye movement; REM, rapid eye movement.
(TIF)

**S2 Fig. *AQP4*-haplotype is associated with a modulation of EEG power in the slow wave range.** Comparison of EEG spectral power across baseline and recovery nights in the slow wave band (0.75–4.5 Hz) within the *AQP4*-haplotype variants HtMa homozygotes (black) and HtMi carriers (red). The *AQP4* HtMi carriers had higher spectral power than the HtMa homozygotes (A; "genotype": F1,121 = 4.2; $P$ < 0.05). The effect was similar in baseline (B) and recovery sleep (C) conditions and confined to the 0.75 to 1.25 Hz band. Inserts in panels B and C represent full NREM sleep spectra for 0 to 20 Hz on log10 scale. Gray shading indicating the slow wave band (S2 Data). Data represents means ± SEM. By-bin unpaired two-tailed $t$ tests; *$P$ < 0.05 (only performed in the slow wave range). AQP4, aquaporin 4; EEG, electroencephalographic; HtMa, Major allele of haplotype; HtMi, Minor allele of haplotype; NREM, non–rapid eye movement.
(TIF)

**S1 Table. Demographics.** Demographic characteristics of the 123 healthy adult volunteers, who participated in one of six 40-hour sleep deprivation studies from the Zürich sleep lab where *AQP4* HtMa homozygotes and HtMi allele carriers were genotyped. No demographic differences between the 2 *AQP4*-haplotype groups was observed, nor was there a difference in haplotype distribution among the 6 studies (Fishers exact $t$ test, $p$ > 0.21). Given the low number of females in the 6 included studies, a potential interaction between *AQP4* haplotype and

gender could not be addressed in the current paper. German versions and validated German translations of questionnaires were used to assess lifestyle and personality traits. Questionnaires included ESS [42] and STAI [43]. Caffeine consumption was estimated based on average caffeine contents per serving (coffee: 100 mg, tea: 30 mg, cola drink: 40 mg [2 dL], energy drink: 80 mg [2 dL], chocolate: 50 mg [100 g]). The APOE genotype, known to modulate Alzheimer disease progression, was evenly distributed among the *AQP4* haplotype groups. *P* values are calculated from students two-tailed *t* tests or Fisher's exact test where appropriate. APOE, Apolipoprotein E; AQP4, aquaporin 4; ESS, Epworth Sleepiness Scale; HtMa, Major allele of haplotype; HtMi, Minor allele of haplotype; STAI, State-Trait Anxiety Inventory. (DOCX)

**S2 Table. Investigated SNPs.** Alleles: presented as [major allele/minor allele]. Position: Position of SNP in gene. Location: Location of single nucleotid polymorphism on chromosome 18 in the genome. Assay ID: Thermofisher Taqman® SNP genotype assay ID nr. $MAF_{predicted}$: MAF predicted by dbSNP analysis tool in a CEU and TSI population (to approximate the Swiss population). $MAF_{study}$: MAF in the entire genotyped study population (*n* = 134), including 2 rare genotypes and 9 elderly subjects excluded from analysis. dbSNP, The Single Nucleotide Polymorphism Database; CEU, Utah Residents from North and West Europe; MAF, minor allele frequency; SNP, single nucleotide polymorphism; TSI, Toscani in Italy; (DOCX)

**S3 Table. Visually scored sleep variables.** Data on the visually scored sleep variables and their modulation by sleep deprivation and the *AQP4* haplotype. Values represent mean ± SEM in baseline and recovery nights for the 2 haplotype groups. Analysis of the recovery nights were restricted to 480 minutes. Sleep efficiency: percentage of total sleep time per 480 min. Stages N1–N3: NREM sleep stages (N3 refers to slow wave sleep). Sleep latency: time from lights-out to the first occurrence of N2 sleep. REM sleep latency: time from sleep onset to the first occurrence of REM sleep. F- and *P* values: two-way mixed-model ANOVA with factors "genotype" (HtMa homozygotes, HtMi carriers), "condition" (baseline, recovery), and their interaction. AQP4, aquaporin 4; HtMa, Major allele of haplotype; HtMi, Minor allele of haplotype; MT, movement time; NREM, non–rapid eye movement; REM, rapid eye movement; TIB, time in bed; TST, total sleep time; WASO, wakefulness after sleep onset. (DOCX)

**S1 Data. Individual genotyping and haplotyping.**
(XLSX)

**S2 Data. Individual whole-night NREM spectral power and spectral energy data.** NREM, non–rapid eye movement.
(XLS)

**S3 Data. Individual NREM spectral power and spectral energy data for the first 4 NREM episodes.** NREM, non–rapid eye movement.
(XLSX)

**S4 Data. Individual data for PVT median speed, PVT lapses, and SSS ratings.** PVT, psychomotor vigilance test; SSS, Stanford sleepiness scale.
(XLSX)

**S5 Data. Individual whole-night REM spectral power and spectral energy data.** REM, rapid eye movement.
(XLSX)

## Acknowledgments

We thank Dr. Marianna Di Chiara, Laura van Bommel, Silke Feil, and Dr. Samuel Koller for their assistance with *AQP4* genotyping, and Dr. Brice Ozenne for his assistance with statistical analysis.

## Author Contributions

**Conceptualization:** Sara Marie Ulv Larsen, Hans-Peter Landolt, Maiken Nedergaard, Sebastian Camillo Holst.

**Data curation:** Sara Marie Ulv Larsen, Hans-Peter Landolt, Wolfgang Berger, Sebastian Camillo Holst.

**Formal analysis:** Sara Marie Ulv Larsen, Hans-Peter Landolt, Sebastian Camillo Holst.

**Funding acquisition:** Sara Marie Ulv Larsen, Hans-Peter Landolt, Maiken Nedergaard, Gitte Moos Knudsen, Sebastian Camillo Holst.

**Investigation:** Sara Marie Ulv Larsen, Hans-Peter Landolt, Gitte Moos Knudsen, Sebastian Camillo Holst.

**Methodology:** Sara Marie Ulv Larsen, Hans-Peter Landolt, Wolfgang Berger, Sebastian Camillo Holst.

**Project administration:** Sara Marie Ulv Larsen, Hans-Peter Landolt, Sebastian Camillo Holst.

**Resources:** Sara Marie Ulv Larsen, Hans-Peter Landolt, Wolfgang Berger, Gitte Moos Knudsen, Sebastian Camillo Holst.

**Supervision:** Hans-Peter Landolt, Wolfgang Berger, Maiken Nedergaard, Gitte Moos Knudsen, Sebastian Camillo Holst.

**Validation:** Sara Marie Ulv Larsen, Hans-Peter Landolt, Sebastian Camillo Holst.

**Visualization:** Sara Marie Ulv Larsen, Maiken Nedergaard, Sebastian Camillo Holst.

**Writing – original draft:** Sara Marie Ulv Larsen, Sebastian Camillo Holst.

**Writing – review & editing:** Sara Marie Ulv Larsen, Hans-Peter Landolt, Wolfgang Berger, Maiken Nedergaard, Gitte Moos Knudsen, Sebastian Camillo Holst.

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
