## [Editor Report · Decision Letter 0]

17 Dec 2019

Dear Dr Holst, 

Thank you for submitting your manuscript entitled "Haplotype of the astrocytic water channel AQP4 modulates slow wave energy in human NREM sleep" for consideration as a Short Report by PLOS Biology.

Your manuscript has now been evaluated by the PLOS Biology editorial staff, as well as by an Academic Editor with relevant expertise, and I am writing to let you know that we would like to send your submission out for external peer review.

However, before we can send your manuscript to reviewers, we would like you to consider including and discussing the following reference: DOI: 10.1111/cns.13194 

In addition, we need you to complete your submission by providing the metadata that is required for full assessment. To this end, please login to Editorial Manager where you will find the paper in the 'Submissions Needing Revisions' folder on your homepage. Please click 'Revise Submission' from the Action Links and complete all additional questions in the submission questionnaire.

Please re-submit your manuscript within two working days, i.e. by Dec 19 2019 11:59PM.

***Please be aware that, due to the voluntary nature of our reviewers and academic editors, manuscripts may be subject to delays due to their limited availability during the holiday season. Please also note that the journal office will be closed entirely 21st- 29th December inclusive, and 1st January 2020. Thank you for your patience.***

Kind regards,

Gabriel Gasque, Ph.D.,

Senior Editor

PLOS Biology

---

## [Decision Letter · Decision Letter 1]

30 Jan 2020

Dear Dr Holst,

Thank you very much for submitting your manuscript "Haplotype of the astrocytic water channel AQP4 modulates slow wave energy in human NREM sleep" for consideration as a Short Report by PLOS Biology. As with all papers reviewed by the journal, yours was evaluated by the PLOS Biology editors as well as by an Academic Editor with relevant expertise and by three independent reviewers. You will note that reviewer 3, Jack A Wells, has signed his comments.

Based on the reviews, we will probably accept this manuscript for publication, assuming that you will modify the manuscript to address the points raised by the reviewers. Please also make sure to address the data and other policy-related requests noted at the end of this email.

We expect to receive your revised manuscript within two weeks. 

Your revisions should address the specific points made by each reviewer. Please note that PLOS does not accept references to data not shown. Thus, as pointed out by reviewers 1, please include those data either in the main body of the manuscript or in the supporting information.

Please submit the following files along with your revised manuscript:

In addition to the remaining revisions and before we will be able to formally accept your manuscript and consider it "in press", we also need to ensure that your article conforms to our guidelines. A member of our team will be in touch shortly with a set of requests. As we can't proceed until these requirements are met, your swift response will help prevent delays to publication.

*Copyediting*

*Published Peer Review History*

*Early Version*

*Submitting Your Revision*

Sincerely,

Gabriel Gasque, Ph.D., 

Senior Editor

PLOS Biology

ETHICS STATEMENT:

-- Please include the ID number of your protocols approved by the ethics committee of the Canton of Zurich for research on human subjects.

-- Please indicate whether the written consent given by participants was informed. 

--Please indicate if you conducted your experiments in accordance to the principles expressed in the Declaration of Helsinki or any other national or international ethical guidelines.

DATA POLICY:

We note that in your Data Availability Statement you wrote: “Data are from the Human Sleep Psychopharmacology Laboratory and are available from the authors at request. Please contact: Prof. Landolt at landolt@pharma.uzh.ch.”

PLOS recognizes that, in some instances, authors may not be able to make their underlying data set publicly available for legal or ethical reasons. Our data policy (https://journals.plos.org/plosbiology/s/data-availability) does not overrule local regulations, legislation or ethical frameworks. Where these frameworks prevent or limit data release, authors must make these limitations clear in the Data Availability Statement.

For more details, please refer to this page: https://journals.plos.org/plosbiology/s/data-availability#loc-acceptable-data-access-restrictions

Please note, however, it is not acceptable for an author to be the sole named individual responsible for ensuring data access.

Also note that we actually do not require all raw data. Rather, we ask for the individual quantitative observations that underlie the data summarized in the figures and results of your paper. For an example see here: http://www.plosbiology.org/article/info%3Adoi%2F10.1371%2Fjournal.pbio.1001908#s5

If you agree it is possible to share these data, they can be made available in one of the following forms:

Regardless of the method selected, please ensure that you provide the individual numerical values that underlie the summary data displayed in the following figure panels: Figures 2A-C, 3AB (including insets), 4A-C, and S1AB.

Please also ensure that the figure legends in your manuscript include information on where the underlying data can be found, and ensure your supplemental data file/s has a legend.

Reviewer remarks:

Reviewer #1: This is an interesting and well-written paper that shows that AQP4 haplotype predicts NREM slow wave energy and response to sleep deprivation. The paper is clear and the analyses are well conducted. I have some comments on the analysis and interpretation. 

- The authors state that there was no difference in other frequency bands, with "data not shown". Since the full spectrum is in the fig. 2 panel inset, could this be replaced with a statement of (p>0.05) instead, or of some other metric of how they determined lack of a difference? This seems like an important point to back up with data, and it appears to me that the data are present.

- The methods describe performing the analyses on both EEG power and EEG energy, but the results only describe the effects for energy, and power is not discussed. Could the authors clarify what the results of their other analysis were or provide rationale for not discussing power? Power is the more commonly used metric, and it's not clear whether this is because it showed no differences, or because there was another reason to focus on the energy metric.

- The authors interpret this phenomenon as a feedback loop from glymphatics to CSF energy. That is certainly possible but it seems possible there are other explanations as well - for example, might this haplotype covary with other genetic determinants of sleep architecture?

- Is it correct and not a typo that sleep was scored in 20 second epochs, rather than typical 30 second epochs described in the cited reference? (this is fine for the reported analysis, but note that the reference states 30 s). 

minor: 

- the analysis relies on the implication that the power analysis and hypothesis were designed a priori to performing genetic analyses, rather than selected post hoc. The methods section confirms that this was a priori but I'd recommend briefly adding that to the main text as well. 

- add units to x-axis of fig 2.b, c

Reviewer #2: This is an interesting study investigating the AQP4 haplotypes and NREM sleep slow wave relationship. I was pleased to have the opportunity to review this work. Hypothesis is clearly stated and backed by earlier research. Study is based on analysis of data pooled from former studies and genotyping for AQP4. The topic is appropriate for this journal and manuscript is well written.

I have some comments on the methods that may impact the clarity of the findings: 

1. How were the subjects recruited? Readers would benefit from a general description of who the subjects were and the basic procedure of recruitment.

2. Both groups have relatively low percentage of females. Why is that? Did the authors consider making an analysis comparing the sexes? 

3. Although small in numbers, seven individuals homozygous for the minor allele were identified, but excluded from the analysis. I think it would be interesting for the reader to have the information about the kinds of results this small group had and a short discussion about a possible dose response effect.

Reviewer #3, Jack A Wells: This study uncovers some interesting new evidence that links various measures of sleep quality (EEG during sleep/awake and participant survey) to genes related to AQP4 in the human brain. 

Given emerging evidence for the interplay between AQP4, sleep, the glymphatic system and measures of neuronal activity from EEG, and the possible role of this interplay in the aetiology of neurodegenerative disease, these data are an interesting development towards better understanding these mechanisms in the human brain. As such, I believe that the data presented in this manuscript would be of interest to the readers of Plos Biology and beyond. The data analysis and statistical approach appears appropriate to support the conclusions made, however, I do not have first hand experience of the analysis of EEG or genetic halotype data and therefore I cannot give detailed and informed feedback on the analytical methods employed here. 

There are some issues in the way the data is contextualized to the current literature that I believe should be addressed prior to possible publication:

The author should avoid unnecessary speculation of possible mechanisms (specifically glymphatic activity) that may mediate or link the correlations that they have observed in their measurements:

e.g: 

'AQP4 HtMi-carriers have a stronger parenchymal CSF flow' - the authors have no evidence for this as CSF flow was not measured - the caveats to this statement should be more clearly stated. 

'our findings support the presence of an innate glymphatic-sleep feedback loop' - again no measurements of glymphatics are made in this study. Although there is evidence in the rodent brain that links glymphatic function to AQP4 expression, changes to AQP4 may also modulate several other physiological changes in the brain such as exchange at the blood brain barrier. The wording should be toned down to acknowledge such uncertainty at the current time..

'and suggests deep NREM sleep, not AQP4, as the main regulator of glymphatic flow' - In my opinion this is overly speculative given the lack of glymphatic measurements in this study and the current uncertainty of the relationship of glymphatic flow with sleep in the human brain. Again the wording should be adjusted to reflect the current experimental evidence. 

'These data suggest that alterations in AQP4-dependent parenchymal CSF flow' - again there are no measures of parenchymal CSF flow made here. I think the authors should be careful to assume that physiologically relevant differences in AQP4 expression closely relate to differences in glymphatic function in the human brain given that the existing evidence that describes this relationship comes from comparing glymphatic flow in a WT and AQP4 knockout mouse (ie. all or nothing) and from correlations in mouse models of ageing or brain pathology (e.g. TBI) where other changes to brain physiology other then AQP4 are present that could also contribute to the correlations observed. 

'our data provides novel 146 evidence for the existence of sleep dependent AQP4-driven CSF pulsations' - the author should more carefully word this statement given they have no measures of CSF pulsations.. firstly it is not known how the differences in AQP4 between the genetic variants explored here may effect glymphatic function in the human brain (the changes may be so subtle as to have minimal effect compared to, vascular pulseation for example and secondly changes in AQP4 expression may affect other physiological process that related to sleep in addition to the glymphatic system (for example classical bulk CSF flow). 

In addition, the authors should avoid claiming evidence for causality when they have evidence for correlation. For example, other genetic changes that are not investigated in this work could be important for the differences in sleep behaviour that is observed. 

e.g. AQP4-haplotype modulates slow waves in NREM sleep

The title and and similar text should be appropriately toned down in acknowledgement of this. 

Introduction: 'facilitates a circulation of nutrients' - to my understanding there have been no studies to demonstrate that the glymphatic system is a physiologically-important carrier of 'nutrients' to the brain. Could the authors clarify what 'nutrients' they are referring to and reference accordingly. 

'is generated by arterial pulsations from the heartbeat and respiration (3,4).': I can see that reference 3 (Maestre) provides evidence that arterial pulsation drives perivascular CSF movement but what is the evidence that respiration is an important mechanism? Reference 4 does not provide this as there is no measurements of glymphatic function in this work. Can the authors clarify where the evidence for respiration being a driver for glymphatic function derives from? 

'glymphatic flow is known to be positively correlated with slow wave production (10)' - an important caveat of this important study is the use of difference anaesthesia protocols to investigate correlations between EEG measures and glymphatic inflow. I think it is important to mention this caveat to readers who may not be aware of this..

'and recently sleep driven parenchymal CSF pulsations in the fourth ventricle were demonstrated in 56 the human brain (13), providing the first evidence for human glymphatic mechanisms'

- Can the authors clarify what a parenchymal CSF pulsation is (and how it is measured)? -I found this hard to understand..

- Can the author clarify exactly how the measurements presented in ref 13 provide evidence for glymphatic mechanisms in the human brain?

---

## [Editor Report · Decision Letter 2]

2 Apr 2020

Dear Dr Holst,

On behalf of my colleagues and the Academic Editor, Paul Shaw, I am pleased to inform you that we will be delighted to publish your Short Reports in PLOS Biology. 

Early Version

PRESS 

Kind regards,

Alice Musson

Publishing Editor 

PLOS Biology

on behalf of

Gabriel Gasque,

Senior Editor

PLOS Biology